# Split and Merge Proxy: pre-training protein-protein contact prediction by mining rich information from monomer data

## Abstract

Protein-protein contact prediction is a key intelligent biology computation technology for complex multimer protein function analysis but still sufferers from low accuracy. An important problem is that the number of training data cannot meet the requirements of deep-learning-based methods due to the expensive cost of capturing structure information of multimer data. In this paper, we solve this data volume bottleneck in a cheap way, borrowing rich information from monomer data. To utilize monomer (single chain) data in this multimer (multiple chains) problem, we propose a simple but effective pre-training method called Split and Merger Proxy (SMP), which utilizes monomer data to construct a proxy task for model pre-training. This proxy task cuts monomer data into two sub-parts, called pseudo multimer, and pre-trains the model to merge them back together by predicting their pseudo contacts. The pre-trained model is then used to initialize for our target – protein-protein contact prediction. Because of the consistency between this proxy task and the final target, the whole method brings a stronger pre-trained model for subsequent fine-tuning, leading to significant performance gains. Extensive experiments validate the effectiveness of our method and show the model performs better than the state of the art by 11.40% and 2.97% on the P@ $L/10$ metric for bounded benchmarks DIPS-Plus and CASP-CAPRI, respectively. Further, the model also achieves almost 1.5 times performance superiority to the state of the art on the harder unbounded benchmark DB5. The code, model, and pre-training data will be released after this paper is accepted.

## 1 Introduction

Proteins are large molecules consisting of amino acids (also called residues) sequences. Protein-protein contact prediction aims to compute the constraints between given protein sequences (specifically whether residue (can be understood as an individual amino acid) on one protein are in contact with residue on the other protein), which is important for the structural or functional analysis of protein complexes. The predicted constraints reveal the relationships between each residue pair of the two protein sequences, which can not only benefit complex protein structure prediction but also be useful for many kinds of protein function analysis scenarios, e. g. developing new drugs and designing new proteins. The success of RaptorX Wang et al. (2017); Xu et al. (2021) and AlphaFold2 Jumper et al. (2021) demonstrates the application potential of deep learning in the computational biology field and inspired a series of new biological computation methods. However, when extending the deep model to protein-protein (inter-chain) contact prediction, recent works have not achieved satisfying performance as the aforementioned successful works do. An important bottleneck is data quantity limitation.

Many well-known successful deep learning systems are almost trained under large-scale datasets. For example, in computer vision (CV), ConvNet Krizhevsky et al. (2012); Simonyan & Zisserman (2014); He et al. (2016)) and ViT Dosovitskiy et al. (2020); Liu et al. (2021); Yuan et al. (2021) are trained on ImageNet Deng et al. (2009) which has 14 million labeled data who provides enough vision category information of real word. For natural language processing (NLP), the most popular language model BERT Devlin et al. (2018) is trained on document-level data BooksCorpus Zhu et al. (2015) and English Wikipedia in an unsupervised manner. And in computational biology, the

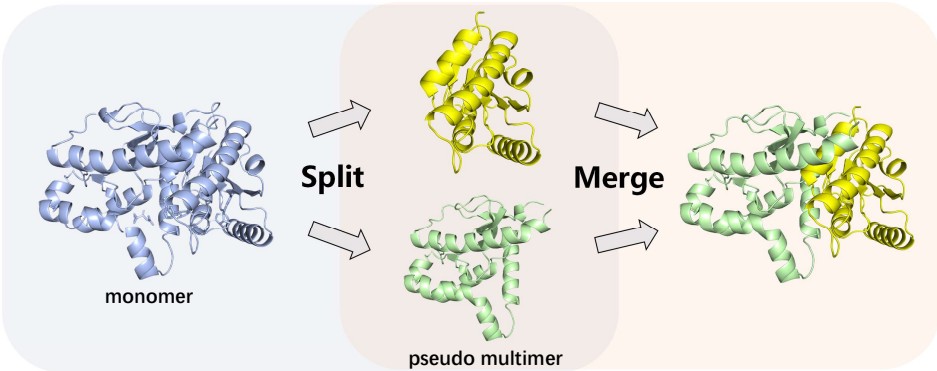

Figure 1: The main idea of Split and Merger Proxy (best viewed in color). In the pre-training stage, a monomer (single chain) is firstly split into two sub-parts that are treated as pseudo multimers (a pair of chains). And then the deep model is pre-trained by learning to merge the pseudo multimers back by predicting their protein-protein contacts.

recent most popular protein structure prediction model AlphaFold2 Jumper et al. (2021) is trained on about 400k monomer data, 60k with 3D structure labels of Protein Data Bank (PDB) wwp (2019) and 350k protein sequence, and achieves electron microscope accuracy. Obviously, existing human-level accurate and successful artificial intelligence models also need big data to train. However, the number of the current largest open-sourced multimer training data Morehead et al. (2022) is much lower than the aforementioned topics, which is only 15k and limits the performance of the deep model. The main reason is the expensive cost of capturing the multimer protein structural information by high-accurate devices. So to tackle the problem of the scarcity of training data, we focus on finding a cheap way to obtain additional data and avoid the extra cost.

Our main idea is to expand the training data by introducing the monomer data into the training step for protein-protein contact prediction. The existing monomer data is free and also can provide useful biological prior. ComplexContact Zeng et al. (2018) is the first work introducing the monomer data into the multimer contact prediction task. It proves the potential value of the monomer data to the multimer task. But obviously, there is an unneglectable task gap between the monomer and the multimer. Specifically, the monomer can only provide information about one chain while the multimer task requires more. So ComplexContact Zeng et al. (2018) suffers from that task gap and existing contact prediction methods often neglect these data. In this paper, we design a novel and effective pre-training method called Split and Merger Proxy (SMP) to introduce monomer data into the protein-protein contact prediction task more effectively, which reduces the aforementioned task gap and leads to better results.

The proposed SMP is a proxy task for contact prediction pre-training. As shown in Figure 1, SMP generates pseudo multimer data from monomers and utilizes that data to pre-train the contact prediction model. In particular, a single protein is **split** into two sub-parts that are treated as a pseudo multimer. That pseudo data are used to train the contact prediction model, equal to guide the model to **merge** these split data back. Although the pseudo multimer data contain biological noise, they can provide additional richer information that complements the existing multimer data. The training targets of SMP and the final task are both contact prediction, so there is no task gap in the fine-tuning stage. The pre-trained model can be fine-tuned on the real multimer data without any modification, leading to a better final model and more accurate contact results.

Our main contributions are as follows:

- We design a novel proxy task, Split and Merger Proxy (SMP), to pre-train contact prediction models on the monomer data more effectively. From the best of our knowledge, this is the first work to leverage the monomer protein data to pre-train the multimer protein contact prediction task.

- Experiments show that we achieve a new state-of-the-art and improve the P@ $L/10$ metric by a large margin – 11.40% and 2.97% respectively on DIPS-Plus and CASP-CAPRI benchmarks when compared with the latest state-of-the-art DeepInteract Morehead et al.

(2022). Moreover, we almost achieve 1.5 times more performance than GeoTrans on the harder unbounded benchmark DB5.

# 2 RELATED WORKS

## 2.1 PROTEIN-PROTEIN CONTACT PREDICTION

Intra-protein contact prediction has been well treated Jumper et al. (2021); Baek et al. (2021), but protein-protein contact prediction has not been extensively studied. Some early works Weigt et al. (2009); Morcos et al. (2011); Ekeberg et al. (2014) used direct-coupling analysis (DCA) to disentangle direct and indirect correlations to infer potential relationships between amino acids at different positions. With the great success of Convolutional Neural Network (CNN) LeCun et al. (1998) in CV area, Zeng et al. (2018); Yan & Huang (2021); Roy et al. (2022) applied CNN to multimer contact prediction. Zeng et al. (2018) used two CNNs, one with 1D convolution processed sequence information and the other with 2D convolution encoded MSA information. Yan & Huang (2021) utilized more biological features (e.g., inter-protein docking pattern, physico-chemical information and sequence conservation) as inputs to the neural network to enrich the information carried by multimer data. Because He et al. (2016) demonstrated that deeper networks could learn more discriminative features from the dataset, Roy et al. (2022) used a deeper dilated residual network Yu et al. (2017) to capture relationships between residues. Due to each protein has 3D structure, Fout et al. (2017); Liu et al. (2020); Morehead et al. (2022); Xie & Xu (2022) designed graph neural network (GNN) Scarselli et al. (2008) to predict contacts between proteins. They first built a graph for each protein, the residue on each protein is regarded as a node, and whether the residues in the protein are connected is regarded as an edge. Fout et al. (2017) used graph convolution Kipf & Welling (2016) to get the graph representation of the underlying protein structure and a fully convolutional network (FCN) was utilized to determine contacts between two proteins. Liu et al. (2020) employed weights sharing GNNs to obtain the residue features of each protein, then they devised multilayer CNNs as the interaction module to perform contact prediction. Based on this, Morehead et al. (2022) designed graph transformers to encode the geometric information in multimers, e.g., the distance and direction between residues and the amide angle. Xie & Xu (2022) believed that simply building the residue graph was not enough, so they built two more graphs, e.g., atom graph and surface graph, then they did message passing in each graph. Since AlphaFold2 Jumper et al. (2021) has achieved surprising results in monomer structure prediction, Evans et al. (2021); Bryant et al. (2022); Gao et al. (2022) extended it to multimer contact prediction. Evans et al. (2021) took into account permutation symmetry, position encoding of different chains in multimer, and multimer multiple sequence alignment (MSA) construction for contact prediction. Bryant et al. (2022); Gao et al. (2022) directly spliced multimer as monomer and fed it into AlphaFold2 to get contact prediction. But due to the small level of existing multimer data, current models are less accurate in protein-protein contact prediction.

## 2.2 PRE-TRAINING IN PROTEIN MODELING

Pre-training from a lot of data can provide a good prior knowledge for the model, so it achieves great success in data science community, such as computer vision and natural language processing areas. Some recent works introduced pre-training paradigm to protein modeling area. Rao et al. (2021); Rives et al. (2021); Elnaggar et al. (2021); Chowdhury et al. (2021); Fang et al. (2022); Lin et al. (2022) used Masked Language Model (MLM) proxy task Devlin et al. (2018) to learn residue embedding from massive protein sequences. Rives et al. (2021); Elnaggar et al. (2021); Chowdhury et al. (2021) directly utilized transformer Vaswani et al. (2017) as pre-training network to capture potential biological patterns of amino acids. Since MSAs can provide a certain biological prior for the model, Fang et al. (2022); Lin et al. (2022) devised the same Evoformer network as AlphaFold2 Jumper et al. (2021) and Rao et al. (2021) designed MSA transformer to fully integrate the MSA information into the transformer architecture in pre-training stage, which can make the network directly learn evolutionary information. Because each atom of the protein in PDB Database wwp (2019) has 3D coordinates, Gligorijević et al. (2021); Chen et al. (2022) designed distance prediction and the dihedral angle prediction proxy tasks, then they got the underlying structural representations for monomers and achieved excellent performance in protein classification tasks. Due to the lack of multimer data and the cost of collecting multimer data is expensive, it is difficult

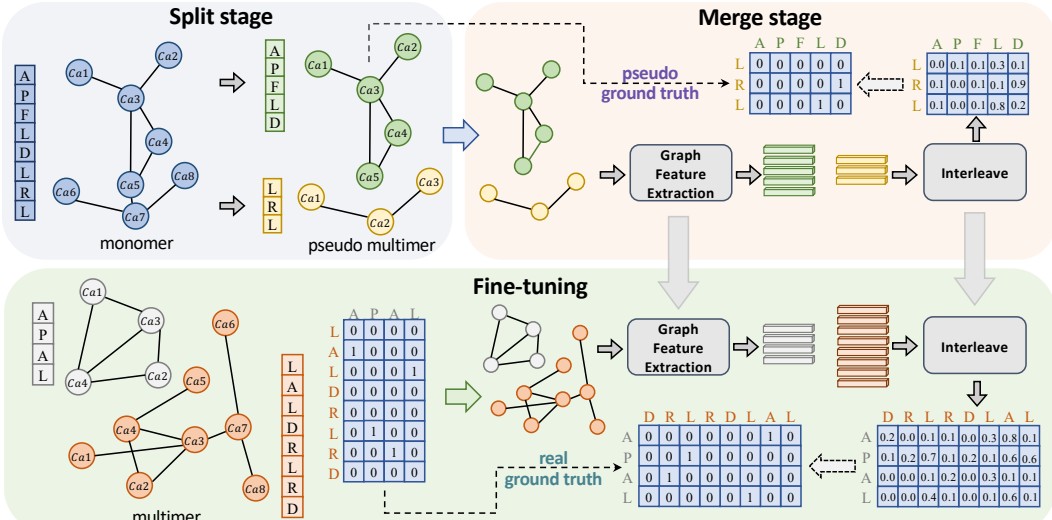

Figure 2: The Framework of the proposed Split and Merge Proxy (SMP) pre-training method (best viewed in color). The split stage cuts a monomer (the sequence "APFLDLRL" in the figure) into a pseudo multimer consisting of two sub-parts (the two sub-parts "APFLD" and "LRL" in the figure) and computes the contact ground truth. The merge step is the pre-training process, which trains the model to predict the contact relationships on the split data, essentially equal to merging the split sequences back. Note that the sequence here is just an example, not the real chain. And in the fine-tuning stage, the whole contact predictor, including the graph feature extractor and the interleave module, is directly fine-tuned without any modification on the real multimer data.

to build effective pre-training on existing multimer data. So in this paper, we design a novel proxy task to adapt the monomer into multimer contact prediction, which can pre-train the model getting stronger performance.

# 3 METHOD

## 3.1 TASK DEFINITION

The protein-protein contact prediction, also called residue-residue interface interaction prediction, aims to compute the contact relationship map $A \in (0,1)^{L_1 \times L_2}$ between the two given protein chains. The element $A_{i,j}$ is 0 or 1, indicating whether the $i$-th residue in one sequence interacts with the $j$-th residue in another sequence or not. The contact prediction models take multiple kinds of biological features as inputs, such as amino acid sequences $s \in \mathbb{P}^L$ ($\mathbb{P}$ is the set of the amino acid, including 20 kinds of amino acids) and residue 3D coordinates $c \in \mathbb{R}^{L \times 3}$ which is essentially the location of the non-hydrogen atoms. The computation pipeline can be defined as:

$$A^{pred} = f(x_1, x_2), \quad \text{where} \quad x_i = \{s_i, c_i\}, \quad i \in \{1, 2\} \tag{1}$$

To achieve this, whatever details of the function $f$, proteins are often regarded as a graph so the residues are treated as graph nodes and processed by graphic models to extract features. After that, an interleave module fuses these node features and measures the similarities between each residue pair to calculate the contact scores. The whole process is the same as the Fine-tuning block in Figure 2.

## 3.2 SPLIT AND MERGER PROXY

The Split and Merger Proxy (SMP) is an effective proxy task proposed to pre-train the contact prediction model. The main pipeline includes a split step and a merge step, shown in Figure 2. Each monomer sequence is cut into two sub-parts to generate the pseudo multimer data. In the merge step, the model learns the contact prediction task directly on the aforementioned split data without any modification. After that, the model would be fine-tuned on real multimer data.

**Split stage:** We use the monomers from the Protein Data Bank (PDB) wwp (2019) dataset because its monomer data including both amino acid sequences $s \in \mathbb{P}^L$ and corresponding 3D coordinates $c \in \mathbb{R}^{L \times 3}$. The target of the split stage is to generate the pseudo multimer that has the same data structure as the real one, including two sub-sequences ($s_1$ and $s_2$) with structural information ($c_1$ and $c_2$) and their corresponding contact ground truth $A$. We first cut the amino acid sequence into two sub-sequences at a random location:

$$s_1 = s[: l], s_2 = s[l :], \tag{2}$$

where $l$ means the random split index uniformly sampled from the range $R$, keeping each cut sequence informative and avoiding too short split results that contain only small amounts of residues. In other words, this split location is around the center of the given sequence.

And for the 3D structure, we do a similar split operation:

$$c_1 = c[: l], c_2 = c[l :]. \tag{3}$$

We do not operate any kind of normalization on these 3D coordinates, keeping their values still in the monomer coordinate system so ground-truth $A$ could be computed by the following formula directly:

$$A_{i,j} = \begin{cases} 1 & D_{i,j} \leq \lambda \\ 0 & D_{i,j} > \lambda \end{cases} \quad , \quad D_{i,j} = ||c_1[i] - c_2[j]||^2, \tag{4}$$

where $\lambda$ is the threshold to determine whether the $i$-th and $j$-th residue pair contact or not. $||\cdot||$ means the Euclidean distance. This process could be interpreted that the ground-truth contact of pseudo multimer is equal to the intra-contact of the original monomer. Based on the steps mentioned above, monomer data is converted to the pseudo multimer in the form of $\{s_1, s_2, c_1, c_2, A\}$.

**Merge stage:** The merge stage is essentially the mimicking learning of the standard contact prediction training. The model learns to predict the $A$ based on the given pseudo multimer inputs $\{s_1, s_2, c_1, c_2\}$.

We first extract and calculate co-evolution, conservation and geometric information for each cut subsequence by Multiple Sequence Alignments (MSA) and Protein Structure and Interaction Analyzer (PSAIA), respectively. And then, these pieces of information combined with the protein sequence and 3D structural information are sent to a weights-sharing graph feature extractor to extract residue features $F_1 \in \mathbb{R}^{L_1 \times C}$ and $F_2 \in \mathbb{R}^{L_2 \times C}$ like Figure 2 shows. Note that the coordinate values in $c_1, c_2$ all belong in the same monomer coordinate system. So they are all treated to the relative distances of residue pairs in each protein sequence to avoid information leakage. After that, an interleave module computes the interaction features $F_I \in \mathbb{R}^{L_1 \times L_2 \times C'}$, which stores the high-level relationship patterns for each residue pair. Finally, a contact prediction head, often a fully convolutional neural network (FCN), predicts a contact map based on that features. For the prediction, we train it as a binary classification task by utilizing the cross-entropy loss function.

**Fine-tuning stage:** The SMP task is the same as the final contact prediction task, both predicting the protein-protein contact maps. So there is not any task gap between this proxy task and fine-tuning. Every module and parameter of the pre-trained model could be re-used in the final model. So, We feed the real multimer data into the pre-trained model and fine-tune the whole model directly.

## 4 EXPERIMENTS

### 4.1 DATASET AND EVALUATION PROTOCOL

In this section, we conduct several experiments on three popular benchmarks DIPS-Plus Morehead et al. (2021), CASP-CAPRI Lensink et al. (2019; 2021) and DB5 Vreven et al. (2015) datasets.

**DIPS-Plus** is latest open-sourced dataset for protein-protein contact prediction. It provides amino acid sequences and residue coordinates for each multimer data. Except for these pieces of basic information, DIPS-Plus also offers additional different types of biological features such as protrusion index and amide plane normal vector, composing much richer information. After filtering extreme data, such as too long, too short sequences and high relative data with other datasets, the DIPS-Plus dataset still has 15,618 and 3,548 protein complexes for training and validation, respectively, which is the recent known largest open-sourced benchmark. For testing, it provides 32 protein complexes consisting of 16 homodimers and 16 heterodimers to evaluate the model's ability to handle samples of different difficulties.

Table 1: The average top-k precision (P@k) and recall (R@k) on DIPS-Plus test dataset (%).

| | 16 (Homo) | | | 16 (Hetero) | | |
|---|---|---|---|---|---|---|
| Method | P@ $L/10$ | P@ $L/5$ | P@ $L/2$ | P@ $L/10$ | P@ $L/5$ | P@ $L/2$ |
| BIPSPI Sanchez-Garcia et al. (2018) | 0 | 0 | - | 2.00 | 2.00 | - |
| DeepHomo Yan & Huang (2021) | 12.00 | 9.00 | - | - | - | - |
| ComplexContact Zeng et al. (2018) | - | - | - | 16.00 | 15.00 | - |
| GCN Morehead et al. (2022) | 20.00 | 18.00 | - | 8.00 | 7.00 | - |
| GeoTrans Morehead et al. (2022) | 25.00 | 23.00 | - | 14.00 | 11.00 | - |
| GeoTrans + SMP | **39.81** | **33.33** | **26.02** | **20.99** | **20.07** | **15.00** |
| | 32 (All Proteins) | | | | | |
| Method | P@ $L/10$ | P@ $L/5$ | P@ $L/2$ | R@ $L$ | R@ $L/2$ | R@ $L/5$ |
| BIPSPI Sanchez-Garcia et al. (2018) | 1.00 | 1.00 | - | 1.00 | 0.40 | 0.30 |
| GCN Morehead et al. (2022) | 16.00 | 12.00 | - | 10.00 | 6.00 | 3.00 |
| GeoTrans Morehead et al. (2022) | 19.00 | 17.00 | - | 15.00 | 9.00 | 4.00 |
| GeoTrans + SMP | **30.40** | **26.70** | **20.51** | **24.00** | **16.02** | **8.56** |

Table 2: The average top-k precision and recall on CASP-CAPRI 13 & 14 dataset.

| | 14 (Homo) | | | 5 (Hetero) | | |
|---|---|---|---|---|---|---|
| Method | P@ $L/10$ | P@ $L/5$ | P@ $L/2$ | P@ $L/10$ | P@ $L/5$ | P@ $L/2$ |
| BIPSPI Sanchez-Garcia et al. (2018) | 0 | 0 | - | 0 | 3.00 | - |
| DeepHomo Yan & Huang (2021) | 2.00 | 2.00 | - | - | - | - |
| ComplexContact Zeng et al. (2018) | - | - | - | 8.00 | 5.00 | - |
| GCN Morehead et al. (2022) | 11.00 | 13.00 | - | 11.00 | 9.00 | - |
| GeoTrans Morehead et al. (2022) | 13.00 | 11.00 | - | 31.00 | **24.00** | - |
| GeoTrans + SMP | **18.63** | **14.37** | **11.57** | **32.00** | 23.49 | **18.35** |
| | 19 (All Proteins) | | | | | |
| Method | P@ $L/10$ | P@ $L/5$ | P@ $L/2$ | R@ $L$ | R@ $L/2$ | R@ $L/5$ |
| BIPSPI Sanchez-Garcia et al. (2018) | 0 | 1.00 | - | 2.00 | 1.00 | 0.1 |
| GCN Morehead et al. (2022) | 10.00 | 9.00 | - | 11.00 | 6.00 | 2.00 |
| GeoTrans Morehead et al. (2022) | 19.00 | 14.00 | - | 13.00 | 8.00 | **4.00** |
| GeoTrans + SMP | **21.97** | **16.77** | **13.36** | 14.33 | 8.34 | 3.91 |

**CASP-CAPRI** has been well known as a biologically joint challenge since 2014, aiming to assess the computational methods of modeling protein structures. Morehead et al. (2022) re-organized the data of the 13th and 14th CASP-CAPRI challenge sessions Lensink et al. (2019; 2021), filtering the overlap between the original CASP-CAPRI data and the DIPS-Plus. These filtered data include 14 homodimers and 5 heterodimers and are used to evaluate the ability of real-world applications and cross-set generalization of models trained on the DIPS-Plus training set.

**DB5** (Docking Benchmarks version 5 Vreven et al. (2015)) is a traditional benchmark dataset for protein-protein contact prediction, including 140 training, 35 evaluation and 55 testing samples. DB5 consists of unbounded protein complexes that have varying contact types. In contrast, complexes in DIPS-Plus and CASP-CAPRI are bounded and their multiple chains are already conformed with each other. So it can indicate the performance and effectiveness of our model on different types of complexes.

**Evaluation** All the experiments follow the standard evaluation protocol in existing multimer contact prediction benchmarks. To assess the accuracy of the prediction, the top-$k$ precision and recall are adopted as the evaluation metrics, where $k \in \{L/30, L/20, L/10, L/5, L/2, L\}$ with $L$ being the length of the shortest chain.

## 4.2 IMPLEMENTATION DETAILS

We generate the pseudo multimer data from all monomers before 2018-4-30 from Protein Data Bank (PDB) wwp (2019). There are $60,206$ pdb files in total. Each file contains sequence and structural information for the protein. Monomers that cannot be parsed by Biopython Cock et al. (2009) (containing unknown atoms; missing atoms; chain numbers are not in order and so on) are filtered out. Except that each protein file contains several conformations, we only keep the first one and abandon the other. We set the split range $R = \{1/3 \sim 2/3\}$ so that the cut position is close to the middle of the given sequence to get pseudo multimers. Too short split proteins whose length of any chain is less than 20 are dropped. The threshold $\lambda$ used to calculate the contact ground truth is set as 6 Å following the same procedure that real multimer utilizes Morehead et al. (2021). Finally, there are $22,589$ pseudo multimers, about 1.5 times of the existing real multimer dataset. Whatever for the pseudo or real multimer data, we also use HHBlits Remmert et al. (2012) with Uniclust30 Mirdita et al. (2017) database for MSA, and PSAIA Mihel et al. (2008) to calculate geometric features.

Our SMP is a pre-training method that is not tightly bound to a specific model. So we combine SMP with the Geometric Transformer (GeoTrans Morehead et al. (2022)) to evaluate the effectiveness of

Table 3: The average top-k precision and recall on DB5 test dataset.

| Method | 55 (Hetero) | | | | | |
|---|---|---|---|---|---|---|
| | P@ $L/10$ | P@ $L/5$ | P@ $L/2$ | R@ $L$ | R@ $L/2$ | R@ $L/5$ |
| BIPSPI Sanchez-Garcia et al. (2018) | 0.20 | 0.10 | - | 0.30 | 0.10 | 0.04 |
| ComplexContact Zeng et al. (2018) | 0.30 | 0.30 | - | 0.70 | 0.30 | 0.10 |
| GCN Morehead et al. (2022) | 0.60 | 0.70 | - | 1.30 | 0.80 | 0.30 |
| GeoTrans Morehead et al. (2022) | 0.90 | 1.10 | - | 1.80 | 1.00 | 0.34 |
| GeoTrans + SMP | **1.78** | **1.88** | **1.55** | **2.53** | **1.45** | **0.69** |

Table 4: SMP vs self-supervised pre-training (SSL) on DIPS-Plus test dataset.

| Row | Model | PreTrain | P@ $L/10$ | P@ $L/5$ | P@ $L/2$ | P@ $L$ | R@ $L$ | R@ $L/2$ | R@ $L/5$ | R@ $L/10$ |
|---|---|---|---|---|---|---|---|---|---|---|
| 1 | GCN | - | 16.00 | 12.00 | - | - | 10.00 | 6.00 | 3.00 | - |
| 2 | GCN | SMP | 18.96 | 15.64 | 11.61 | 8.24 | 13.58 | 10.04 | 5.36 | 3.14 |
| 3 | GeoTrans | - | 19.00 | 17.00 | - | - | 15.00 | 9.00 | 4.00 | - |
| 4 | GeoTrans | SSL | 20.87 | 18.19 | 14.62 | 12.40 | 17.46 | 9.88 | 4.87 | 2.83 |
| 5 | GeoTrans | SMP | **30.40** | **26.70** | **20.51** | **15.87** | **24.00** | **16.02** | **8.56** | **4.79** |

SMP in the following experiments. The batch size of pre-training and fine-tuning are all set as 48 (except the fine-tuning one of CASP-CAPRI is set as 32 because of the cross-domain evaluation setting of CASP-CAPRI). Other experimental settings, including loss function, optimizer, learning rate and so on, are all kept the same to the latest open-sourced state-of-the-art GeoTrans.

## 4.3 COMPARISON WITH STATE-OF-THE-ART METHODS

We compare several state-of-the-art multimer contact prediction methods including BIP-SPI Sanchez-Garcia et al. (2018), ComplexContact Zeng et al. (2018), DeepHomo Yan & Huang (2021), GCN Morehead et al. (2022) and GeoTrans Morehead et al. (2022). Except that the input of ComplexContact is the amino acid sequence, the other methods take both amino acid sequence and 3D structural information as inputs, which are the same as our model.

Table 1 shows the comparison results between SMP and other methods on the DIPS-Plus dataset, demonstrating that SMP outperforms existing state-of-the-art method GeoTrans Morehead et al. (2022) by a large margin. For homologous complexes, SMP outperforms GeoTrans by 7.43% on the harder metric P@ $L/2$ and even 18.36% on P@ $L/20$, demonstrating that SMP can learn more useful residue representation and contact prediction knowledge from additional pseudo multimer data. For more difficult heterologous complexes, SMP also surpasses GeoTrans 4.49% on harder P@ $L/2$. These heterologous performances benefit from the potential consistency with the pseudo multimer and heterologous proteins. Specifically, the cut chains usually have low sequence identities, sharing certain similar properties and distributions of the real heterologous data, making SMP an obvious improvement on heterologous multimers. From an overall perspective, the proposed SMP brings significant gains compared with GeoTrans by 11.4% at P@ $L/10$ and 8.00% at R @$L$, proving that our SMP brings more discriminative expression for multimer contact prediction whatever homologous or heterologous complexes.

Tables 2 presents the average top-$k$ metrics of SMP on the CASP-CAPRI dataset, specifically, 19 challenging protein complexes (14 homodimers and 5 heterodimers). SMP also surpasses the state-of-the-art method GeoTrans on P@ $L/10$ by 5.63% on 14 homologous when keeping comparable performances for 5 heterologous. SMP achieves improvements for several different settings, demonstrating that the pre-training of SMP learns many valuable patterns of contact prediction from pseudo multimers to help learn real multimer prediction effectively.

On the DB5 dataset in Table 3, SMP also exceeds the precision of GeoTrans for all metrics. All methods perform poorly due to testing hard and unseen unbound complexes with varying contact types that are not necessarily conformal. However, SMP still shows more than 1.5 times better performance than GeoTrans in almost all metrics. It indicates that SMP has good cross-domain capabilities and has the potential to be used in real-world applications of complex contact prediction.

Overall, this pre-training paradigm plays a considerable role in various types of downstream protein-protein contact prediction tasks (cross set and unbound set), showing good robustness with SMP.

## 4.4 ABLATION STUDIES

### 4.4.1 COMPARISON WITH DIFFERENT PRE-TRAINING PARADIGM AND CONTACT PREDICTOR

Previous comparisons show the effectiveness of the combination of our SMP with the latest state-of-the-art model GeoTrans Morehead et al. (2022). In this ablation study, we further investigate the

Table 5: Partial pre-training results on DIPS-Plus test dataset.

| Row | Ratio | P@ $L/10$ | P@ $L/5$ | P@ $L/2$ | P@ $L$ | R@ $L$ | R@ $L/2$ | R@ $L/5$ | R@ $L/10$ |
|---|---|---|---|---|---|---|---|---|---|
| 1 | 0 | 19.00 | 17.00 | - | - | 15.00 | 9.00 | 4.00 | - |
| 2 | 1/5 | 18.22 | 15.58 | 13.35 | 11.06 | 16.80 | 10.48 | 4.84 | 2.76 |
| 3 | 1/4 | 18.61 | 19.02 | 15.10 | 11.76 | 17.08 | 11.14 | 5.68 | 2.90 |
| 4 | 1/3 | 24.64 | 21.36 | 16.59 | 11.92 | 16.93 | 12.08 | 6.14 | 3.35 |
| 5 | 1/2 | 26.20 | 21.29 | 15.85 | 12.77 | 18.09 | 11.50 | 6.40 | 3.84 |
| 6 | 1 | **30.40** | **26.70** | **20.51** | **15.87** | **24.00** | **16.02** | **8.56** | **4.79** |

Table 6: Partial fine-tuning results on DIPS-Plus test dataset.

| Row | Ratio | P@ $L/10$ | P@ $L/5$ | P@ $L/2$ | P@ $L$ | R@ $L$ | R@ $L/2$ | R@ $L/5$ | R@ $L/10$ |
|---|---|---|---|---|---|---|---|---|---|
| 1 | 0 | 5.98 | 3.70 | 2.08 | 1.98 | 1.78 | 0.82 | 0.62 | 0.56 |
| 2 | 1/5 | 15.42 | 15.40 | 12.25 | 9.63 | 14.02 | 8.59 | 4.20 | 1.96 |
| 3 | 1/4 | 19.83 | 16.12 | 12.53 | 10.80 | 15.87 | 9.71 | 4.91 | 3.10 |
| 4 | 1/3 | 19.84 | 17.11 | 12.96 | 10.62 | 14.29 | 8.70 | 4.56 | 2.50 |
| 5 | 1/2 | 23.99 | 19.55 | 15.49 | 11.92 | 16.97 | 11.11 | 5.53 | 3.22 |
| 6 | 1 | **30.40** | **26.70** | **20.51** | **15.87** | **24.00** | **16.02** | **8.56** | **4.79** |

superiority of our SMP. We combine SMP with different contact predictor to prove its generalization and also compare the SMP with other pre-training method to show the advantage of the SMP design. All related results are shown in Table 4.

To investigate the influence of the combined contact predictor with SMP, we change the graph feature transfer module from Transformer into the Graph Convolutional Network (GCN) Kipf & Welling (2016). This GCN only has a total of 33k parameters, which is quite much lower than the 1.4m parameters of the Transformer one. So this setting can show the generalization of the SMP on a small-scale model. From the 1st and 2nd lines of Table 4, it can be seen that our SMP still brings a 3.64% performance increase under the P@ $L/5$. It indicates that the SMP paradigm keeps strong generalization on the small-scale model, showing the potential for extensions of future different types and levels of contact predictors.

To show the superiority of the SMP design, we construct another pre-training method by adapting a popular self-supervised learning (SSL) paradigm Proteinbert Brandes et al. (2022) to train on the monomer data with 3D structural cues. This method pre-trains the model by a self-supervised proxy task that guides the model to reconstruct the inputs by partial observation, providing different pre-training mechanisms compared with our SMP. As shown in the 3rd ∼ 4th lines of Table 4, this SSL method provides average 1% gains on all metrics, proving the fact that monomers bring much useful information for this multimer task from a different view. But when compared with the SMP (5th line), SMP still shows stronger performance and outperforms the SSL by 5.89% on the harder metric P@ $L/2$, indicating the superiority of the SMP design that can utilize information in 3D structures more effectively and further eliminate the task gap between the pre-training and fine-tuning.

### 4.4.2 PARTIAL PRE-TRAINING RESULTS

We study the effectiveness of pre-training data volume for SMP and conduct partial pre-training experiments with different degrees of monomer data. We set five partial pre-training ratios $\{1/5, 1/4, 1/3, 1/2, 1\}$. When comparing the 2nd line of Table 5 with the 1st line (without pre-training), we find that the performance has some fluctuation when the number of introduced pseudo multimers is small. This is caused by the biological noise introduced by the small-scale pseudo data, which is eliminated when the scale increases and clearly indicated in Table 5 3rd∼6th lines. Obviously, when the amount of pre-trained data reaches $1/4$ (in the 3rd line), SMP has introduced certain precision and recall gains except on P@ $L/10$ metric than GeoTrans (in the 1st line), with an average improvement of 2%. Moreover, as the amount of pre-trained data increases, the performance gradually improves, proving that SMP guides the model to learn rich contact prediction to provide beneficial initialization parameters for contact prediction models.

### 4.4.3 PARTIAL FINE-TUNING RESULTS

The pre-trained model has the potential to achieve satisfying performances only trained with small-scale training data. So we aim to explore the effect of SMP for fine-tuning with different scale data. We use six partial fine-tuning ratios, which belong to the set $\{0, 1/5, 1/4, 1/3, 1/2, 1\}$. The 1st line of Table 6 shows that SMP surpasses the traditional method BIPSPI (Table 3) without any fine-tuning, which indicates that pseudo multimer can provide prior knowledge that is relevant to the real multimers contact prediction. Moreover, the 3rd line of Table 6 shows that our model achieves

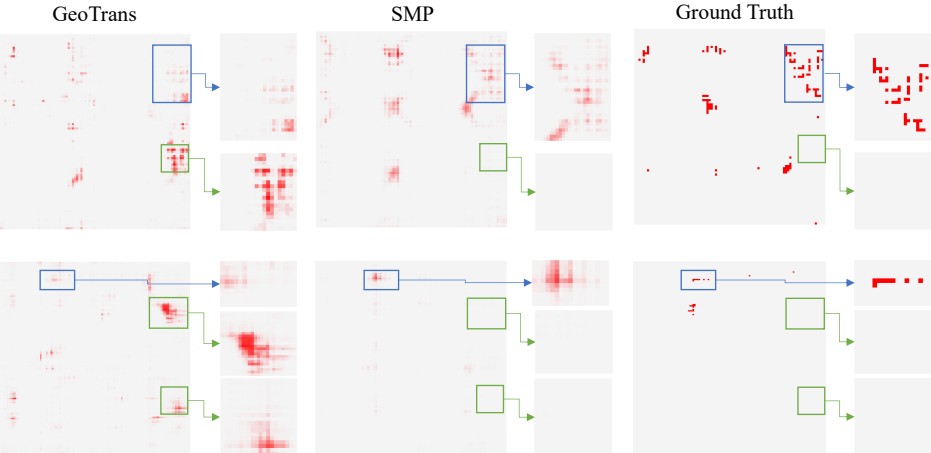

GeoTrans                    SMP                    Ground Truth

Figure 3: Contact visualization results of the 2 multimer 4LIW (first row) and 4DR5 (second row). The GeoTrans's predictions, SMP's predictions and ground truths are corresponding to the left, middle and right columns, respectively.

Table 7: Different split ranges results on DIPS-Plus test dataset.

| Row | Range | P@ $L/10$ | P@ $L/5$ | P@ $L/2$ | P@ $L$ | R@ $L$ | R@ $L/2$ | R@ $L/5$ | R@ $L/10$ |
|---|---|---|---|---|---|---|---|---|---|
| 1 | $2/5 \sim 3/5$ | 21.94 | 19.47 | 14.37 | 11.66 | 17.63 | 11.43 | 6.39 | 3.65 |
| 2 | $1/3 \sim 2/3$ | **30.40** | **26.70** | **20.51** | **15.87** | **24.00** | **16.02** | **8.56** | **4.79** |
| 3 | $1/4 \sim 3/4$ | 23.11 | 20.00 | 17.22 | 14.35 | 20.80 | 12.80 | 5.85 | 3.35 |
| 4 | $1/5 \sim 4/5$ | 23.90 | 20.75 | 14.17 | 11.36 | 16.84 | 10.71 | 6.16 | 3.40 |

comparable results to our combined predictor GeoTrans Morehead et al. (2022) only with 1/4 training data demonstrating that the SMP pre-training can provide knowledge that can be effectively re-used and transferred to the real multimer scenario. With further increasing the data volume in the 4th $\sim$ 6th line of Table 6, it can finally achieve 30.4% on metric P@ $L/10$, surpassing the previous state-of-the-art method GeoTrans. These experiments prove that our pre-training paradigm can effectively reduce the dependence on real data and make the model adapt to different volume-level training data situations, having the potential to save the extra cost of collecting multimer data.

#### 4.4.4 DIFFERENT SPLIT RANGE RESULTS

We study the influence of different split ranges on the split stage for SMP to find the optimal one. We set four split intervals settings $\{2/5 \sim 3/5, 1/3 \sim 2/3, 1/4 \sim 3/4, 1/5 \sim 4/5\}$. As shown in the 2nd line of Table 7, we find the performance is best when the split interval is $1/3 \sim 2/3$. This appropriate range makes the random interval relatively close to the middle of the protein chain and avoids yielding one of the monomers with a too short length simultaneously.

### 4.5 VISUALIZATION

We also visualize some prediction results of GeoTrans in Figure 3. We exhibit a homologous multimer (i.e., PDB ID: 4LIW) and a heterologous multimer (i.e., 4DR5) from the DIPS-Plus test set. The blue box in Figure 3 indicates that SMP successfully can predict several positive contacts that GeoTrans neglects. And the green box in Figure 3 shows that our SMP can eliminate some false positives provided by GeoTrans. All these bounded areas demonstrate that SMP is more accurate in multimer contact prediction than the state-of-the-art method GeoTrans, demonstrating that the model pre-trained by SMP can carry several types of new advantages over the original one.

## 5 CONCLUSION

This paper introduces the Split and Merger Proxy (SMP), a simple yet effective pre-training framework for protein-protein contact prediction to solve the limited number of multimers by using rich monomer information. SMP splits monomer data into pseudo multimers and trains the model to merge them back together by predicting its pseudo contact interaction, which reduces the task gap between this proxy task and the final target, leading to significant performance gain. It demonstrates that splitting monomers benefit multimer contact prediction tasks and also implies that monomers data may have the potential for other downstream computational multimer protein tasks.

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

Table 8: Different batch size results on three benchmarks. DS: dataset, BZ: batch size.

| DS | BZ | P@ $L/10$ | P@ $L/5$ | P@ $L/2$ | P@ $L$ | R@ $L$ | R@ $L/2$ | R@ $L/5$ | R@ $L/10$ |
|---|---|---|---|---|---|---|---|---|---|
| | bs=32 | 30.28 | 26.63 | 19.64 | 15.09 | 20.85 | 14.19 | 7.31 | 4.01 |
| DIPS-Plus | bs=48 | 30.40 | 26.70 | 20.51 | 15.87 | 24.00 | 16.02 | 8.56 | 4.79 |
| | bs=72 | 28.29 | 24.34 | 19.93 | 15.32 | 21.02 | 13.84 | 6.59 | 3.70 |
| | bs=32 | 23.52 | 18.94 | 13.78 | 11.17 | 12.58 | 7.96 | 4.35 | 2.52 |
| CASP-CAPRI | bs=48 | 23.36 | 18.84 | 13.92 | 11.14 | 13.88 | 8.64 | 4.33 | 2.57 |
| | bs=72 | 16.28 | 14.91 | 12.90 | 10.02 | 14.34 | 9.15 | 4.83 | 2.68 |
| | bs=32 | 1.59 | 1.69 | 2.24 | 2.05 | 3.96 | 2.04 | 0.56 | 0.22 |
| DB5 | bs=48 | 1.78 | 1.88 | 1.55 | 1.33 | 2.53 | 1.45 | 0.69 | 0.33 |
| | bs=72 | 1.42 | 1.84 | 1.60 | 1.65 | 2.96 | 1.46 | 0.69 | 0.24 |

# A   APPENDIX

## A.1   DETAILED NETWORK DESIGN

We use the same network architecture as GenoTrans Morehead et al. (2022), the details are as follows, a k-nearest neighbor graph Preparata & Shamos (2012) is applied to construct a graph of each chain, the residue on each chain is regarded as a node, and the first k residues with the smallest distance from other residues are considered for connecting with the current residue, the k is 20. For network architecture, we use the 2 layers of graph transformer Dwivedi & Bresson (2020) with batch normalization Ioffe & Szegedy (2015), each transformer has 4 attention head and 128 hidden size to get rich node and edge representations. We also use a 14 layers of dilated residual network Yu et al. (2017) for interleave module. It contains a 4 residual block, and each block is composed of 2D convolution with kernel size $3 \times 3$ and instance normalization. A squeeze-and-excitation (SE) attention Hu et al. (2018) is added after each block to capture channel-wise information. In addition, we use Adam optimizer Kingma & Ba (2014) with the learning rate of $1e^{-3}$, the weight decay rate of $1e^{-2}$ and the batch size of 48 to train the network. The dropout Srivastava et al. (2014) of 0.2 and an early-stopping patience period of 5 epochs are applied to avoid network over-fitting. Since there are more non-contact sites than contacts in the protein-protein contact prediction task, there is a huge class imbalance, and we use weighted cross entropy with a positive class weight of 5 to overcome this imbalance. To experiment on unbounded dataset DB5, we fine-tune models on 140 training and 35 validation complexes of DB5 with the learning rate of $1e^{-5}$.

## A.2   BATCH SIZE TUNING

The benchmarks that we utilize for fine-tunning and evaluating our model cover several contact prediction scenarios, including bounded Morehead et al. (2021), cross-set Lensink et al. (2019; 2021) and unbounded situations Vreven et al. (2015). So the hyper-parameters in fine-tuning may affect the final results. In this sub-section, we aim to find the optimal batch size on these downstream datasets respectively. We set different batch size settings of 32, 48 and 72 on all utilized benchmarks as shown in Table 8. We find that when pre-training with batch size of 48 achieving the best result of 30.40 % on P@ $L/10$ metric on the bounded benchmark DIPS-Plus. On the cross-set benckmark CASP-CAPRI, fine-tuning with batch size of 32 is optimal and can obtain 23.52 % on P@ $L/10$. For the unbounded DB5 dataset, it achieves the best result of 1.78 % on P@ $L/10$ metric with batch size 48.

## A.3   COMPARISON WITH DIFFERENT GRAPH PRE-TRAINING METHODS

Since GeoTrans is a graph-based model, we use different graph pre-training methods to compare with SMP to show the superiority of the SMP design. We compare several simple graph pre-training methods including Mask node Hu et al. (2020), Mask edge Hu et al. (2020), and PHD Li et al. (2021). Table 9 shows the contrast results between SMP and other graph pre-training methods. We find that all these graph pre-training methods have improved to the baseline method GeoTrans, and SMP outperforms other methods by a large margin in all metrics. This experiment indicates that SMP has no task gap between the pre-training stage and the fine-tuning stage in the protein-protein contact prediction task, but other graph pre-training methods have.

Table 9: Comparison with different graph pre-training methods on DIPS-Plus test dataset.

| Row | Model | P@ $L/10$ | P@ $L/5$ | P@ $L/2$ | R@ $L$ | R@ $L/2$ | R@ $L/5$ |
|---|---|---|---|---|---|---|---|
| 1 | GeoTrans Morehead et al. (2022) | 19.00 | 17.00 | - | 15.00 | 9.00 | 4.00 |
| 2 | GeoTrans + Mask Node Hu et al. (2020) | 20.87 | 18.19 | 14.62 | 17.46 | 9.88 | 4.87 |
| 3 | GeoTrans + Mask Edge Hu et al. (2020) | 20.26 | 17.67 | 14.31 | 16.37 | 10.47 | 5.16 |
| 4 | GeoTrans + PHD Li et al. (2021) | 20.33 | 17.32 | 14.50 | 16.51 | 10.75 | 5.34 |
| 5 | GeoTrans + SMP | **30.40** | **26.70** | **20.51** | **24.00** | **16.02** | **8.56** |

Table 10: Comparison with different protein language methods on DIPS-Plus test dataset.

| Row | Model | P@ $L/10$ | P@ $L/5$ | P@ $L/2$ | R@ $L$ | R@ $L/2$ | R@ $L/5$ |
|---|---|---|---|---|---|---|---|
| 1 | GeoTrans Morehead et al. (2022) | 19.00 | 17.00 | - | 15.00 | 9.00 | 4.00 |
| 2 | ESM-1b Rives et al. (2021) | 18.96 | 16.97 | 14.26 | 12.85 | 8.44 | 3.98 |
| 3 | ESM-MSA-1b Rao et al. (2021) | 23.49 | 20.68 | 16.86 | 14.73 | 9.71 | 4.89 |
| 4 | GeoTrans + SMP | **30.40** | **26.70** | **20.51** | **24.00** | **16.02** | **8.56** |

## A.4 COMPARISON WITH DIFFERENT PROTEIN LANGUAGE MODELS

The large protein language model has the potential to learn prior biological knowledge from massive monomer sequences. We investigate large-scale protein language models ESM-1b Rives et al. (2021) and ESM-MSA-1b Rao et al. (2021) on DIPS-Plus test dataset as shown in Table 10. It demonstrates that SMP has higher accuracy than other pretrained protein language models for all metrics. Both ESM-1b (650M parameters) and ESM-MSA-1b (100M parameters) have much more parameters than our SMP (4.5M parameters), reflecting the effectiveness and light weight of our SMP. Since ESM-1b does not use any MSA and geometric information, its accuracy is lower than the baseline method GeoTrans. ESM-MSA-1b acquires a strong biological prior knowledge to get higher accuracy. The above experiments demonstrate that SMP splits the monomer into the pseudo multimer and pre-training the pseudo multimer could be more beneficial than using monomer pre-training directly.

