# OpenReview forum: "Split and Merge Proxy: pre-training protein-protein contact prediction by mining rich information from monomer data"
_ICLR.cc/2023/Conference — Submitted to ICLR 2023_

### Official Review · Reviewer_oqAy · 2022-10-23

**Confidence:** 5
**Correctness:** 1
**Technical Novelty And Significance:** 3
**Empirical Novelty And Significance:** 3
**Recommendation:** 6

**Clarity, Quality, Novelty And Reproducibility:**

The paper source code is available and the experiment details are provided carefully in the paper. The results should be able easy to reproduce.

The paper was well written and all the technical points are described clearly.

Although this is a simple pretraining approach, the idea is original and interesting. I have more concern on the evaluation methods (please see my comments on weaknesses).

**Strength And Weaknesses:**

Strengths

Interesting idea of creating pretraining data.

The improvement over the SOTA-supervised learning approach is impressive.


Weaknesses

Potential leakage happens
Since the SMP model was pretrained on the PDB dataset. There might be potential leakage happens if the test set of DIPS-Plus, DB5, and CASP-CAPRI  overlap with the data used for pretraining. Even though the objective of SMP which focuses on reconstructing the contacts of broken monomer and the objective of a multimer are different, seeing test instances during pretraining steps provides an unfair advantage over other methods that do not see the test data distribution during their learning. I suggest the authors check for overlapping instances between PDB and  DIPS-Plus, DB5, and CASP-CAPRI and remove them before pretraining to avoid leakage.

Comparison with pretrained protein models
Even the authors have compared their methods to the pretrained language models ProteinBert with the results reported in Table 4, I would suggest the authors compare their approaches to the following pretrained models available at https://github.com/facebookresearch/esm:

+ ESM-1b: this is a standard language model trained on the large database of  protein sequences, demonstrated very good results on contact prediction

+ ESM-MSA-1b: leveraging MSA information for predicting better contact

This ESM pretrained model family was trained with very large data and demonstrated SOTA results on contact prediction.

Comparison with pretrained monomer contact prediction
To demonstrate that it is necessary to break the monomer into two chains and predict the merged contact with the Split and Merger proxy, the following experiments need to be done on the PDB sequences with 3D structure:

+ Take the representation from ESM-1b as the representation of the protein sequence.

+ Predict the contact matrix constructed by looking at the Euclidean distance between residues coordinates provided in the protein 3D data

+ Use the new representation of ESM-1b pretrained on the monomer contact prediction for GeoTrans to further fine-tune for multimer contact prediction.



**Summary Of The Paper:**

Summary of the paper
The paper proposes a new method for pretraining models for protein-protein contact prediction. The authors created pretraining data by collecting sequences and 3D data from the PDB database.

They break a monomer (a single chain) into a multimer with two chains at a random location in the sequence and train a model to predict the contacts of these two chains. The pseudo-contact ground truth between pseudo multimers is defined using the Euclidean distance between residue coordinates provided within the 3D data (residues within 5 Angstrom are considered as having a contact).

The pre-trained models are then fine-tuned further on protein-protein contact prediction datasets using supervised training approaches.

The authors demonstrated that although this is a simple pretraining approach, it provides a significant improvement over SOTA-supervised methods and the other methods that use pre-trained models for protein representation using language models such as the ProteinBert.





**Summary Of The Review:**

Summary of the comments
This is an interesting work that has a good impact on the application domain.
My main concerns are:

+  potential leakage, please fix this and report new results, if the new results are still significant, I am happy to change my score

+ additional comparisons when replacing ProteinBert with ESM-1b, ESM-MSA-1b are needed because these models demonstrated SOTA results on supervised/unsupervised contact prediction.

+ additional comparisons with pretraining monomer contact prediction using PDB data and ESM-1b as back-bond features for proteins are needed to demonstrate that the Split and Merge Proxy is needed.

---

> ### Author Response · Authors · 2022-11-18
> **Response to Reviewer oqAy (1)**
>
> We are grateful to the reviewer for the constructive and insightful comments. Below we provide a point-by-point reply.
>
> Q1. Potential leakage happens Since the SMP model was pretrained on the PDB dataset. There might be potential leakage happens if the test set of DIPS-Plus, DB5, and CASP-CAPRI overlap with the data used for pretraining.
>
> >We check that there is no overlapping instance between our pretrained monomer data and training/test multimer data:
> >
> >>1. The pretrained data is the monomer and the training/test data is the multimer. So under the physical view, there is no overlap.
> >>2. Both pretrained monomer and training/test multimer are all from the PDB dataset[1]. The monomer and multimer in PDB correspond to different ID numbers, so there is no overlap between pretrained dataset and DIPS-Plus, DB5, and CASP-CAPRI test set.
> PDB IDs of each dataset are provided: <https://anonymous.4open.science/r/pdb_ids>.
>
> [1] Protein Data Bank: the single global archive for 3D macromolecular structure data[J]. Nucleic acids research, 2019, 47(D1): D520-D528.
>
> Q2. Additional comparisons when replacing ProteinBert with ESM-1b, ESM-MSA-1b.
>
> >1. Sorry to make that confusion. We do not use ProteinBert as the representation of protein sequences. The fourth row of Table 4 in the original paper shows that we use a self-supervised strategy similar to ProteinBert to pre-training the parts of the graph model in GeoTrans.
> >
> >2. We also use pretraind protein language model ESM-1b and ESM-MSA-1b as the representation of the protein sequence and train the GeoTrans. The results are as follows:
> |            |         |        |     DIPS-Plus     |       |        |        |
> |:----------:|:-------:|:------:|:-----------------:|:-----:|:------:|:------:|
> |            |         |        | 32 (All proteins) |       |        |        |
> |   Method   | P@ L/10 | P@ L/5 |       P@ L/2      |  R@ L | R@ L/2 | R@ L/5 |
> |  GeoTrans  |  19.00  |  17.00 |         -         | 15.00 |  9.00  |  4.00  |
> |   ESM-1b   |  18.96  |  16.97 |       14.26       | 12.85 |  8.44  |  3.98  |
> | ESM-MSA-1b |  23.49  |  20.68 |       16.86       | 14.73 |  9.71  |  4.89  |
> |     SMP    |  30.40  |  26.70 |       20.51       | 24.00 |  16.02 |  8.56  |
>
> >It demonstrates that SMP has higher accuracy than other pretrained protein language models for all metrics. Both ESM-1b (650M parameters) and ESM-MSA-1b (100M parameters) have much more parameters than our SMP (4.5M parameters), reflecting the effectiveness and light weight of our SMP. Since ESM-1b does not use any MSA and geometric information, its accuracy is lower than the baseline method GeoTrans. ESM-MSA-1b acquires a strong prior biological knowledge to get higher accuracy.  All these experiments have been added to the appendix due to the time, and we will reorganize the experiments in the text in the official version.

---

> > ### Author Response · Authors · 2022-11-18
> > **Response to Reviewer oqAy (2)**
> >
> > Q3. Additional comparisons with pretraining monomer contact prediction using PDB data and ESM-1b as back-bond features for proteins.
> >
> > > We also compare the pre-training ESM-1b for monomer contact prediction. Since there is no open-source code, we reproduce it by ourselves. The detail is as follows: the Adam optimizer with the learning rate of $1e^{-5}$, the weight decay rate of $1e^{-2}$, and the batch size of $2$ to pre-training the ESM-1b on monomer contact prediction. We train $10$ epochs. The performance (top@ L/10) for the monomer contact prediction is increased from $24.75$ to $72.05$, showing it has been trained well on that pre-training proxy task. And then, we use pre-training ESM-1b as backbone features for protein sequences and train the GeoTrans. The results are as follows:
> > |                     |         |        |     DIPS-Plus     |       |        |        |
> > |:-------------------:|:-------:|:------:|:-----------------:|:-----:|:------:|:------:|
> > |                     |         |        | 32 (All proteins) |       |        |        |
> > |        Method       | P@ L/10 | P@ L/5 |       P@ L/2      |  R@ L | R@ L/2 | R@ L/5 |
> > | pretraining  ESM-1b |  11.71  |  12.69 |       11.50       | 11.51 |  7.04  |  3.25  |
> > |         SMP         |  30.40  |  26.70 |       20.51       | 24.00 |  16.02 |  8.56  |
> >
> > >It seems that the pretraining ESM-1b achieves worse results than the original ESM-1b, also lower than our SMP. We conclude the reasons as follows:
> > > ESM-1b has 650M parameters, which is easily overfitting to the specific monomer contact prediction proxy task. Also, this proxy task has a gap with the final multimer task. So pretraining ESM-1b on that proxy task breaks the good generalization of the original ESM-1b, leading to sub-optimal results. However, the proxy task of our SMP is similar to the final multimer contact prediction. Although there is some noise under our pseudo multimer data, our model only has 4.5M parameters, preventing our model appear overfitting on that noise in pretraining and achieving good results.

---

### Official Review · Reviewer_HQ2g · 2022-10-24

**Confidence:** 3
**Correctness:** 4
**Technical Novelty And Significance:** 3
**Empirical Novelty And Significance:** 3
**Recommendation:** 5

**Clarity, Quality, Novelty And Reproducibility:**

The paper is rather clearly written. In terms of quality and novelty it seems to me to be ok, if the focus is restricted to reasoning over proteins. Code is provided.


**Strength And Weaknesses:**

(+) The paper is mostly straightforward and concise, the approach is reasonable.

(+) The evaluation is extensive, contains various datasets and models, ablation studies, and more detailed analysis.

(-) For someone coming from ML some parts are a bit unclear. For example:
p.1 "the constraints between given protein sequences" - what exactly is the form of these constraints?

(-) Related work:
- Overall the paper focuses heavily on the application. Pretraining has become a huge field for graph neural networks in very short time, and the paper is missing related work from there completely, which could be applied to the monomers. At a venue such as ICLR, that work should be mentioned. For example:
Pairwise Half-graph Discrimination: A Simple Graph-level Self-supervised Strategy for Pre-training Graph Neural Networks, IJCAI 21
- "ComplexContact Zeng et al. (2018) is the first work introducing the monomer data into the multimer contact prediction task." - This seems to be an important related work, how exactly do they use the data?

(-) The results seem nearly a bit too good and I wonder what is the issue. It seems that the other models do not use pre-training, therefore I wonder why the authors do not compare to ComplexContact in more metrics, and don't apply some existing GNN pre-training approaches.


**Summary Of The Paper:**

The paper proposes a new approach for protein-protein contact prediction that is based on monomer data (this has been used in related work) and particularly cuts monomers into two sub-parts  and pre-trains the model to merge them back. The results show huge performance gains compared to related works.


**Summary Of The Review:**

Altogether, to me this seems to be a solid contribution with some issues that could be corrected. I am no expert for reasoning over proteins, therefore other reviewers would have to confirm if the claims w.r.t. novelty and SOTA are true.

My main concern is that the work is very application-focused and the paper does not really represent new, more general ML technology. But this seems to be a more general question, how such papers are to be handled. I didn't find guidance for this from ICLR. I would find a more general consideration of GNN pretraining more insightful for the community.

---

> ### Author Response · Authors · 2022-11-18
> **Response to Reviewer HQ2g (1)**
>
> We are grateful to the reviewer for the constructive and insightful comments. Below we provide a point-by-point reply.
>
> Q1. For someone coming from ML some parts are a bit unclear. For example: p.1 "the constraints between given protein sequences" - what exactly is the form of these constraints?
>
> >Sorry to make that confusion. 'Constraint' here refers to a constraint on the geometric relationship between two proteins, specifically whether residues (which can be understood as an individual amino acid) on one protein are in contact with residues on the other protein. So it is exactly equal to the contact map that we predict. We will elaborate more clearly in the final version, especially more reader-friendly for ML background researchers.
>
> Q2. Pretraining has become a huge field for graph neural networks in very short time, and the paper is missing related work from there completely, which could be applied to the monomers. At a venue such as ICLR, that work should be mentioned. For example: Pairwise Half-graph Discrimination: A Simple Graph-level Self-supervised Strategy for Pre-training Graph Neural Networks, IJCAI 21.
> >Thanks for your suggestion. Here we provide some additional comparison with other graph pre-training, including the Mask Node, Mask edge and PHD on the DIPS-Plus dataset. Mask Node means learning to recover the masked nodes of the graph. Mask Edge aims to reconstruct the masked edges of the graph. PHD is your recommended paper. The results are shown in the following table:
> |          |         |        |    DIPS-Plus    |       |        |        |
> |:-----------------:|:-------:|:------:|:------:|:-----:|:------:|:------:|
> |  |         |        |  32 (All proteins)      |       |        |        |
> |       Method      | P@ L/10 | P@ L/5 | P@ L/2 |  R@ L | R@ L/2 | R@ L/5 |
> |      GeoTrans     |  19.00  |  17.00 |    -   | 15.00 |  9.00  |  4.00  |
> |     Mask Node     |  20.87  |  18.19 |  14.62 | 17.46 |  9.88  |  4.87  |
> |     Mask Edge     |  20.26  |  17.67 |  14.31 | 16.37 |  10.47 |  5.16  |
> |        PHD        |  20.33  |  17.32 |  14.50 | 16.51 |  10.75 |  5.34  |
> |        SMP        |  30.40  |  26.70 |  20.51 | 24.00 |  16.02 |  8.56  |
>
> > It indicates that SMP still has much higher accuracy than other GNN pre-training methods in all metrics. All these compared general graph pre-training methods introduce gains. But they do not consider the task gap, so they do not perform better than our SMP. All these experiments have been added to the appendix due to the time, and we will reorganize the experiments in the text in the official version.
>
> Q3. "ComplexContact Zeng et al. (2018) is the first work introducing the monomer data into the multimer contact prediction task." - This seems to be an important related work, how exactly do they use the data?
>
> > They directly train a model on monomers by monomer contact prediction. And then, they directly use that model to predict multimer contact prediction. When facing the multimer task, the multimer is concatenated into a single chain and fed into the model as the monomer does.
> >
> > The main differences between our SMP comparing with ComplexContact are:
> > 1. **Model**: ComplexContact is essentially a monomer task model, so its model design has some important drawbacks that limit its multimer task performances. For example, the order of concatenating two sequences into a multimer substantially affects the performance, which is not considered when training ComplexContact. But our SMP is essentially a model designed for the multimer, so it does not face this kind of drawbacks.
> > 2. **Method**: ComplexContact merges two monomers into a single one, but our SMP splits one monomer into two monomers (i.e., a multimer).
> > 3. **Conclusion**: ComplexContact shows that the monomer system can be transferred to the multimer task. SMP aims to demonstrate that monomers can be used to expand data for the multimer task.
> >
> Q4. I wonder why the authors do not compare to ComplexContact in more metrics.
> > ComplexContact does not release open-source code and data now, so we follow the results reported in GeoTrans [1] on the same test sets DIPS-Plus, CASP-CAPRI, and DB5. We will try to build more comparisons with that method under more metrics. Thanks for your suggestion.
>
> [1] Morehead A, Chen C, Cheng J. Geometric Transformers for Protein Interface Contact Prediction[C]. ICLR, 2022.

---

> > ### Author Response · Authors · 2022-11-18
> > **Response to Reviewer HQ2g (2)**
> >
> > Q5. My main concern is that the work is very application-focused and the paper does not really represent new, more general ML technology. But this seems to be a more general question, how such papers are to be handled. I didn't find guidance for this from ICLR. I would find a more general consideration of GNN pretraining more insightful for the community.
> > > Effectively utilizing existing training samples is an important problem in ML when there is insufficient data, which is a common ML problem. This paper shows that we can leverage one kind of dataset (monomer data) for another kind of task (multimer prediction), whose concepts and insights can be transferred to a general multimer protein processing technology, which also has the potential to inspire the ML community in related tasks.
> > >

---

### Official Review · Reviewer_bLda · 2022-10-26

**Confidence:** 4
**Correctness:** 3
**Technical Novelty And Significance:** 3
**Empirical Novelty And Significance:** 3
**Recommendation:** 6

**Clarity, Quality, Novelty And Reproducibility:**

The paper is well-organized and clearly written. The approach is novel but the impact may be limited to protein-protein contact prediction.
Question:
What other domains/applications might the concept of SMP be applied to?


**Strength And Weaknesses:**

Strength:
The proposed SMP approach is a smart and novel trick to pretrain models without the use of labeled protein-protein contact dataset.
Strong empirical results are shown with proposed approach.

Weaknesses:
Impact seems limited to protein-protein contact prediction


**Summary Of The Paper:**

The paper proposes a novel self-supervised approach to pretrain models on protein-protein contact predictions. The approach, called Split and Merger Proxy (SMP), uses protein monomer samples by splitting them into two sub-parts as a proxy to protein-protein complex and pre-trains the model to predict its original ‘merged’ form. Through this approach, the models can break away from the limitations of the small dataset of labeled protein-protein contact. Experiments have shown that SMP improves the performance of current baselines to achieve state-of-the-art results in downstream protein-protein contact prediction.

**Summary Of The Review:**

The proposed approach is novel and a clever way to exploit the relatively larger dataset of protein monomers. The empirical performance gains are impressive. However, the impact might be limited to only protein-protein contact prediction.

---

> ### Author Response · Authors · 2022-11-18
> **Response to Reviewer bLda**
>
> We are grateful to the reviewer for the constructive and insightful comments. Below we provide a point-by-point reply.
>
> Q1. Impact seems limited to protein-protein contact prediction. What other domains/applications might the concept of SMP be applied to?
> > Thanks for your suggestion, we will try to extend our method into other kinds of multimer tasks, e. g. multimer structure prediction [1,2] protein-protein docking [3, 4] and protein-protein interaction [5, 6], in the future. Conceptly, SMP can be utilized for most of the multimer tasks because split monomers can always be seen as pseudo multimers. This kind of data-level modification can be easily adapted to that topics.
> > Besides, protein-protein contact prediction itself can also benefit other topics, such as drug discovery [7, 8]. For example, it can indicate information on the Antigen-antibody interaction, which can simplify the pipeline of drug discovery. So the success of our SMP in the contact prediction also has potential to promote other related application scenarios in other topics.
>
> [1] Evans R, O’Neill M, Pritzel A, et al. Protein complex prediction with AlphaFold-Multimer[J]. BioRxiv, 2022: 2021.10. 04.463034.
>
> [2] Ko J, Lee J. Can AlphaFold2 predict protein-peptide complex structures accurately?[J]. BioRxiv, 2021.
>
> [3] Ganea O E, Huang X, Bunne C, et al. Independent se (3)-equivariant models for end-to-end rigid protein docking[C]. ICLR, 2022.
>
> [4] Ghani U, Desta I, Jindal A, et al. Improved docking of protein models by a combination of alphafold2 and cluspro[J]. BioRxiv, 2022: 2021.09. 07.459290.
>
> [5] Rodrigues C H M, Myung Y, Pires D E V, et al. mCSM-PPI2: predicting the effects of mutations on protein–protein interactions[J]. Nucleic acids research, 2019, 47(W1): W338-W344.
>
> [6] Bryant P, Pozzati G, Elofsson A. Improved prediction of protein-protein interactions using AlphaFold2[J]. Nature communications, 2022, 13(1): 1-11.
>
> [7] Zhou G, Gao Z, Ding Q, et al. Uni-Mol: A Universal 3D Molecular Representation Learning Framework[J]. 2022.
>
> [8] Raybould M I J, Marks C, Krawczyk K, et al. Five computational developability guidelines for therapeutic antibody profiling[J]. Proceedings of the National Academy of Sciences, 2019, 116(10): 4025-4030.

---

### Official Review · Reviewer_2Wyz · 2022-11-04

**Confidence:** 4
**Correctness:** 3
**Technical Novelty And Significance:** 2
**Empirical Novelty And Significance:** 3
**Recommendation:** 5

**Clarity, Quality, Novelty And Reproducibility:**

The simple idea of the paper is clearly articulated.  Given the success of AlphaFold at predicting multimers it is not surprising that it works.
However, I would not describe the idea as particularly novel, since AlphaFold is trained on monomers as well.

**Strength And Weaknesses:**

Strengths:
* The chief benefit of the suggested approach is it's simplicity.
* The comparison of the performance of GeoTrans with and without SMP is good ablation for checking the extent to which the method improves performance.

Weaknesses:
* Discussion of prior work.  RoseTTAFold2 is explicitly geared towards interaction prediction, as is AF2 Multimer.  While these methods are primarily built for structure prediction (rather than phrased as for contact prediction) they may easily used for contact prediction.  For example using the predicted alignment error (PAE) output by AF2.
*  Predicted alignment error as been shown to be effective in predicting existence and strength of contacts in the context of binder design (Bennet et al. 2022).   For RosettaFold a simple option could be pLDDT at the interacting residues.
* How does pre-training compare to relying on generalization from structure prediction with large (e.g. 384 residue crops), as in AlphaFold?
* An empirical comparison to AF2 or RosettaFold is crucial.  I will not consider changing my score if a comparison is not to be added.

In some cases, it seems that the pre-training strategy does not help (in particular, in table 2 in some comparisons to GeoTrans).  Could the authors comment on the situations in which they expect the pre-training will have the largest / smallest improvements?


Nit:
* Check for typos: e.g. suppresses —> surpasses?
* What do you mean by “rich” information in the title?  Would be helpful to clarify this vague language in the main text or choose alternative phrasing.


References:
Bennett, Nathaniel, et al. "Improving de novo protein binder design with deep learning." bioRxiv (2022).
Baek, Minkyung, et al. "Accurate prediction of protein structures and interactions using a three-track neural network." Science 373.6557 (2021): 871-876.

**Summary Of The Paper:**

The paper describes simple pre-training strategy for multi-mer contact prediction.  The strategy is shown to provide empirical benefit across several benchmark tasks.  However the paper does not compare to some of the most widely used methods for predicting contacts (AlphaFold and RosettaFold).

**Summary Of The Review:**

A clear explanation and empirical demonstration of simple idea, but with insufficient demonstration of the novelty and comparison + discussion of existing work (namely AlphaFold and RosettaFold).

---

> ### Author Response · Authors · 2022-11-18
> **Response to Reviewer 2Wyz (1)**
>
> We are grateful to the reviewer for the constructive and insightful comments. Below we provide a point-by-point reply.
>
> Q1 and Q2: Q1. Discussion of prior work. RoseTTAFold2 is explicitly geared towards interaction prediction, as is AF2 Multimer. While these methods are primarily built for structure prediction (rather than phrased as for contact prediction) they may easily used for contact prediction. For example, using the predicted alignment error (PAE) output by AF2. // Q2. Predicted alignment error as been shown to be effective in predicting existence and strength of contacts in the context of binder design [1]. For RosettaFold a simple option could be pLDDT at the interacting residues.
>
> >**Can AlphaFold solve the contact prediction well?**
>
> >Though RoseTTAFold2 and AF2 can indeed be utilized to predict the contact map, a series of experiments below show that our baseline GeoTrans has better accuracy than them on this topic. For the AF2, we evaluate it on the DIPS-PLUS, CASP-CAPRI, and DB5 datasets. And for the AF-multimer, we exclude the DIPS-Plus and DB5 benchmarks because the DIPS-Plus test set is 100% (32/32) overlapped with the training set of multimers in the AF-multimer and DB5 test set is 94.5% (52/55) overlapped with the training set of AF-Multimer while the CASP-CAPRI is not. The results are shown in the following Tables:
> |         |         |        |   Tab.1 DIPS-Plus      |             |        |        |
> |:-----------------:|:-------:|:------:|:------:|:-----------:|:------:|:------:|
> |         |     16 (Homo)     |        |        |  |       16 (Hetero)  |        |
> |       Method      | P@ L/10 | P@ L/5 | P@ L/2 |   P@ L/10   | P@ L/5 | P@ L/2 |
> |        AF2        |  12.78  |  10.67 |  8.72  |     0.27    |  0.62  |  0.43  |
> |        SMP        |  39.81  |  33.33 |  26.02 |    20.99    |  20.07 |  15.00 |
> |  |         |        |     32 (All proteins)   |             |        |        |
> |       Method      | P@ L/10 | P@ L/5 | P@ L/2 |     R@ L    | R@ L/2 | R@ L/5 |
> |        AF2        |   6.53  |  5.64  |  4.57  |     1.92    |  1.40  |  0.63  |
> |        SMP        |  30.40  |  26.70 |  20.51 |    24.00    |  16.02 |  8.56  |
>
> >|        |         |        |   Tab.2  CASP-CAPRI     |            |        |        |
> |:-----------------:|:-------:|:------:|:------:|:----------:|:------:|:------:|
> |          |     14 (Homo)    |        |        | |      5 (Hetero)   |        |
> |       Method      | P@ L/10 | P@ L/5 | P@ L/2 |   P@ L/10  | P@ L/5 | P@ L/2 |
> |        AF2        |   6.34  |  3.70  |  2.32  |     0.0    |   0.0  |  0.77  |
> |    AF-Multimer    |  14.06  |  7.43  |  3.86  |     0.0    |   0.0  |   0.0  |
> |        SMP        |  18.63  |  14.37 |  11.57 |    32.00   |  23.49 |  18.35 |
> | |         |        |    19 (All proteins)     |            |        |        |
> |       Method      | P@ L/10 | P@ L/5 | P@ L/2 |    R@ L    | R@ L/2 | R@ L/5 |
> |        AF2        |   4.67  |  2.73  |  1.91  |    1.24    |  1.18  |  0.99  |
> |    AF-Multimer    |  10.36  |  5.47  |  2.85  |    2.47    |  2.31  |  2.19  |
> |        SMP        |  21.97  |  16.77 |  13.36 |    14.33   |  8.34  |  3.91  |
>
> >|         |         |        |   Tab.3 DB5     |       |        |        |
> |:-----------:|:-------:|:------:|:------:|:-----:|:------:|:------:|
> |  |         |        |   55 (Hetero)     |       |        |        |
> |    Method   | P@ L/10 | P@ L/5 | P@ L/2 |  R@ L | R@ L/2 | R@ L/5 |
> |     AF2     |  0.074  |  0.074 |  0.054 |  0.17 |  0.063 |  0.034 |
> |     SMP     |   1.78  |  1.88  |  1.55  |  2.53 |  1.45  |  0.69  |
>
> >The experimental results above show that AF-multimer can achieve better results than AF2. But our SMP (even our baseline GeoTrans) performs much better than both AF2 and AF-multimer. The reason is that :
> >1. AF2 is a model designed for monomers, so its multimer-treating performance is not quite satisfying.
> >2. AF-multimer can handle multimer structure prediction, but its model design is the same as the AF2. Although it has some technology for multimer applications, its model design still limits its performance to a certain degree.
> >
> >Our SMP is essentially a model designed for the multimer, provided by our baseline GeoTrans. So it does not face this kind of drawbacks.
>
> [1] Bennett N, Coventry B, Goreshnik I, et al. Improving de novo protein binder design with deep learning[J]. bioRxiv, 2022.

---

> > ### Author Response · Authors · 2022-11-18
> > **Response to Reviewer 2Wyz (2)**
> >
> > >**About utilizing PAE and pLDDT in the contact prediction**
> >
> > >From our knowledge, PAE and pLDDT cannot be utilized to tackle the contact prediction topic:
> > >>1. From the original definition, PAE cannot be directly utilized to get the contact prediction results because it is a metric that is computed between the predicted 3D structure and the **ground-truth** 3D structure that cannot be acquired in real inference scenarios.
> > >>2. According to the Reviewer's description of PAE, we guess an additional meaning is utilizing PAE as a distance for the two sequences in a multimer to compute the contact map. This implementation also has problems. (1) PAE can only be computed when the two sequences share the same lengths, but sequences have different lengths in most cases of multimer contact prediction. (2) PAE computes the **relative** errors/distances for each residue pair (According to the definition of [2], PAE is computed in the **relative coordinate system**). However, the contact prediction task needs to compute the **absolute** distances of residue pairs in two sequences (According to the definition of [3], contact or not is computed by the absolute Eculeadan distance between different residues in the **same coordinate system**). Those relative errors/distances cannot be transferred to absolute results.
> > >>3. pLDDT also cannot be utilized for contact prediction directly. pLDDT reflects structure prediction confidences for each residue, which cannot indicate the relationships between a residue pair.
> >
> > Q3. Typos:
> > >Thanks for your suggestion. All typos and unclear descriptions have been updated in the paper.
> >
> > Q4. How does pre-training compare to relying on generalization from structure prediction with large (e.g. 384 residue crops), as in AlphaFold
> > >We conduct experiments on large proteins (Residue number > 384) in DIPS-Plus, DB5, and CASP-CAPRI datasets. The results are shown as follows, demonstrating the good generalization ability of our SMP on large proteins:
> >
> > >| |         |        |   Tab.4  DIPS-Plus      |       |        |        |
> > |:--------------------------------:|:-------:|:------:|:------:|:-----:|:------:|:------:|
> > |              Method              | P@ L/10 | P@ L/5 | P@ L/2 |  R@ L | R@ L/2 | R@ L/5 |
> > |             GeoTrans             |  21.97  |  18.64 |  17.32 | 22.61 |  15.10 |  6.43  |
> > |                SMP               |  32.95  |  32.16 |  24.76 | 31.26 |  22.16 |  11.46 |
> >
> > >|   |         |        |  Tab.5 CASP-CAPRI      |       |        |        |
> > |:---------------------------------:|:-------:|:------:|:------:|:-----:|:------:|:------:|
> > |               Method              | P@ L/10 | P@ L/5 | P@ L/2 |  R@ L | R@ L/2 | R@ L/5 |
> > |              GeoTrans             |   8.08  |  6.73  |  5.34  |  9.04 |  6.09  |  3.53  |
> > |                SMP                |  10.56  |  8.52  |  7.21  | 13.23 |  7.69  |  3.42  |
> >
> > >|  |         |        |    Tab.6 DB5    |      |        |        |
> > |:--------------------------:|:-------:|:------:|:------:|:----:|:------:|:------:|
> > |           Method           | P@ L/10 | P@ L/5 | P@ L/2 | R@ L | R@ L/2 | R@ L/5 |
> > |          GeoTrans          |   0.99  |  1.12  |  1.12  | 1.81 |  0.96  |  0.39  |
> > |             SMP            |   1.67  |  1.94  |  1.73  | 3.01 |  1.62  |  0.76  |
> >
> > Q5. In some cases, it seems that the pre-training strategy does not help (in particular, in table 2 in some comparisons to GeoTrans). Could the authors comment on the situations in which they expect the pre-training will have the largest / smallest improvements?
> > >In Table 1 in the original paper, we can find that the average improvements on heterodimers and homodimers on DIPS-Plus are similar, about 50%. And in Table 3 of the original paper, our method achieves satisfying gains on the DB5 dataset that is entirely heterodimers.
> > >For the CASP-CAPRI results shown in Table 2, the improvement on heterodimers is small. The main reason is that the heterodimer samples in that CASP-CAPRI dataset are harder samples selected by [4]. Except that, evaluation on CASP-CAPRI is essentially the cross-set setting because this dataset does not provide a training set to fine-tune the model. So it is more challenging for the pretrained model. Similar poor results can be found in the aforementioned Table of AlphaFold. All AlphaFold methods only achieve 0 performance on the heterodimers of the CASP-CAPRI dataset, demonstrating it is extremely challenging.
> >
> > [2] Varadi M, Anyango S, Deshpande M, et al. AlphaFold Protein Structure Database: massively expanding the structural coverage of protein-sequence space with high-accuracy models[J]. Nucleic acids research, 2022, 50(D1): D439-D444.
> >
> > [3] Yan Y, Huang S Y. Accurate prediction of inter-protein residue–residue contacts for homo-oligomeric protein complexes[J]. Briefings in bioinformatics, 2021, 22(5): bbab038.
> >
> > [4] Morehead A, Chen C, Cheng J. Geometric Transformers for Protein Interface Contact Prediction[C]. ICLR, 2022.

---

> > > ### Author Response · Authors · 2022-11-18
> > > **Response to Reviewer 2Wyz (3)**
> > >
> > > Q6. Given the success of AlphaFold at predicting multimers it is not surprising that it works. However, I would not describe the idea as particularly novel, since AlphaFold is trained on monomers as well
> > >
> > > >AlphaFold includes AlphaFold2 and AlphaFold-Multimer. Only the former one is trained on the monomer data.
> > > >1. For AlphaFold2, it aims to tackle the **MONOMER** structure prediction. So it is intuitive to train AlphaFold2 on monomers. Although this model shows some ability to tackle the multimer tasks, its performance is much lower than ours, as shown in Table 1-3.
> > > >2. For the AlphaFold-multimer, which is designed to tackle the **MULTIMER** structure prediction, it is well-known that this model provides better results for the multimer tasks than AlphaFold2 [5]. But this model is only trained on multimers without any monomer and also suffers from the data limitation mentioned in our paper. Its results are shown in Table 2. It can be seen that the performance of the AlphaFold-multimer is much lower than ours.
> > > >3. Our SMP aims to tackle the **MULTIMER** contact prediction, but gains benefit from our designed monomer pre-training method. It provides an efficient pre-training method to leverage monomers for this multimer task well and achieves much better results than the AlphaFold2 trained on monomers and AlphaFold-multimer trained only with multimers. The key of our SMP is to show that monomer data can benefit the multimer tasks, which is validated by experimental results. The novelty of our SMP is complementary to the novelty of AlphaFold2, and AlphaFold-Multimer.
> > >
> > > Q7. What do you mean by “rich” information in the title? Would be helpful to clarify this vague language in the main text or choose alternative phrasing.
> > >
> > > > Sorry to make you confusion and thanks for your suggestion. From our results, the performance gains show that our model essentially learns additional patterns compared with our baseline. These patterns are the socalled “rich” information.
> > >
> > > [5] Jumper J, Evans R, Pritzel A, et al. Highly accurate protein structure prediction with AlphaFold[J]. Nature, 2021, 596(7873): 583-589.

---

> > ### Comment · Reviewer_2Wyz · 2022-11-19
> > **Follow-up on Reply**
> >
> > Thank you for your detailed reply -- I have increased my score.

---

> > > ### Author Response · Authors · 2022-11-19
> > > **To reviewer 2Wyz**
> > >
> > > Dear reviewer 2Wyz: Thank you for your reply.

---

### Decision · Program_Chairs · 2023-01-20

**Decision:**

Reject

**Justification For Why Not Higher Score:**

See the above-mentioned weaknesses.

**Justification For Why Not Lower Score:**

N/A

**Metareview: Summary, Strengths And Weaknesses:**

The paper introduces a novel pre-training strategy, called Split and Merger Proxy (SMP), for multimer contact prediction. The approach uses protein monomer samples by splitting them into two sub-parts as a proxy of a protein complex and pre-trains the model to predict its original ‘merged’ form. Through this approach, the models can break away from the limitations of the small dataset of labeled protein-protein contact. The pre-trained models are fine-tuned on protein-protein contact prediction datasets using supervised training approaches.

**Strengths:**
* Reviewers appreciate the simplicity of the new approach. The SMP approach is a smart and novel trick to pretrain models without using labeled protein contact datasets.
*  Strong empirical results are shown. Experiments have shown that SMP improves the performance of current baselines to achieve state-of-the-art results in downstream protein-protein contact prediction. SMP provides empirical benefits across several benchmark tasks. The empirical performance gains are impressive.
* Reviewers agree that although this is a simple pretraining approach, it significantly improves SoTA-supervised methods and other methods that use pre-trained models for protein representation using language models such as the ProteinBert.

**Weaknesses:**
* The initial paper did not compare SMP to widely used methods for predicting contacts (AlphaFold and RosettaFold). The authors performed some additional experiments in the rebuttal to empirically compare the new method to AF2 and RosettaFold.
* Concerns were raised regarding potential leakage through the PDB dataset. The authors investigated the issues, and the paper would benefit from additional confirmatory analyses.
* Reviewers noted limited discussion of prior work with certain key research areas omitted from the Related work section.
* The impact is limited to protein-protein contact prediction. It would be valuable if authors considered other downstream tasks to demonstrate the utility of SMP.
* Additional ablation studies must be performed, comparing SMP to popular pretrained protein models and pretrained monomer contact prediction models.

This paper needs further improvement, and I suggest the authors address criticisms raised by the reviewers and refine the manuscript for a more solid publication.